# LncRNAs Regulate Vasculogenic Mimicry in Human Cancers

**DOI:** 10.3390/cells14080616

**Published:** 2025-04-20

**Authors:** Eloísa Ibarra-Sierra, Mercedes Bermúdez, Carlos Esteban Villegas-Mercado, Macrina B. Silva-Cázares, César López-Camarillo

**Affiliations:** 1Instituto Estatal de Cancerología “Dr. Arturo Beltrán Ortega”, Acapulco Guerrero 39530, Mexico; comite.etica@cancerologiagro.gob.mx; 2Facultad de Odontología, Universidad Autónoma de Chihuahua, Chihuahua 31000, Mexico; mbermudez@uach.mx (M.B.); cmercado@uach.mx (C.E.V.-M.); 3Unidad Académica Multidisciplinaria Región Altiplano, Universidad Autónoma de San Luis Potosí, Matehuala 78700, Mexico; macrina.silva@uaslp.mx; 4Posgrado en Ciencias Genómicas, Universidad Autónoma de la Ciudad de México, Ciudad de México 03100, Mexico

**Keywords:** LncRNAs, microRNAs, vasculogenic mimicry, coregulation networks, cancer

## Abstract

Vasculogenic mimicry (VM) has recently been discovered as an alternative mechanism for nourishing cancer cells in vivo. During VM, tumor cells align and organize themselves into three-dimensional (3D) channel-like structures to transport nutrients and oxygen to the internal layers of tumors. This mechanism mainly occurs in aggressive solid tumors and has been associated with poor prognosis in oncologic patients. Long non-coding RNAs (lncRNAs) are essential regulators of protein-encoding genes involved in cancer development and progression. These single-stranded RNA molecules regulate critical cellular functions in cancer cells including cell proliferation, apoptosis, angiogenesis, VM, therapy response, migration, invasion, and metastasis. Recently, high-throughput RNA-sequencing technologies have identified thousands of lncRNAs, but only a small percentage of them have been functionally characterized in human cancers. The vast amount of data about its genomic expression in tumors can allow us to dissect their functions in cancer biology and make them suitable biomarkers for cancer diagnosis and prognosis. In this study, we reviewed the current knowledge about the role of lncRNAs in regulating VM in cancer. We also examined the molecular mechanisms of lncRNAs and highlight several commonalities in the cellular functions associated with VM between diverse cancer types. Future directions for research focused on deciphering their function in VM are delineated. Finally, the potential of selected lncRNAs as novel therapeutic targets in RNA-based molecular interventions is also discussed.

## 1. Introduction

An efficient distribution of nutrients and oxygen supply is required for cancer cell nourishment and tumor growth. Recent studies indicate that tumor cells grow in three-dimensional (3D) architectures, where gradients for nutrients and oxygen uptake are formed [1]. Indeed, cancer cells in the external layers of tumors show high proliferation rates, whereas the internal core mainly present necrotic cells. Therefore, to maintain a concentration gradient of O_2_ and nutrients and provide novel routes for metabolic waste, cancer cells may alternatively align and reorganize themselves to form 3D channel-like structures in vivo, a phenomenon termed vasculogenic mimicry (VM) [2]. The 3D channels representative of VM are functional as they transport red blood cells and nutrients, providing an alternative fuel source to support tumor nourishment. VM has been detected in almost all solid human tumors, conferring aggressive behavior, higher metastatic potential, and resistance to therapy, resulting in a poor prognosis for oncologic patients [3]. New blood vessel formation via endothelial cells from pre-existing vasculature (angiogenesis) and VM can function in a coordinated fashion, resulting in mosaic vessels featuring a mixture of connected VM channels and blood vessels [4]. Mechanistically, VM is activated by proteins involved in the response to hypoxia orchestrated by the hypoxia-induced factor 1 alpha (HIF-1α) transcription factor and the VEGF-α, Wnt, EphA2, FAK, PI3K, AKT, Notch, and TGF-SMAD signaling pathways [5].

Although it has been reported that non-coding RNAs, such as microRNAs (miRNAs), may regulate VM in diverse cancer types [5], the role of long non-coding RNAs (lncRNAs) in modulating VM remains poorly understood, with very few reports on the topic. LncRNAs are a novel class of single-strand RNAs defined as transcripts of about 200 nucleotides in their mature form, transcribed from thousands of genetic loci dispersed across the human genome [6]. Two early concepts regarding its evolutionary origin and functions have been demystified: (i) while the concept of RNAs lacking protein-coding potential has been widely accepted, recent studies have shown that lncRNAs may contain small open reading frames (less than 100 codons in length) that are pervasively translated into mini peptides [7]; and (ii) lncRNAs were mainly considered as “junk” genetic material, an old concept in genomics [8], but have now been widely accepted as non-coding RNAs, which are transcribed in a regulated manner, modulating gene expression and phenotypes in human cancers. Nevertheless, lncRNAs are predominantly non-coding RNAs that exhibit critical roles in regulating gene expression in a pathological and physiological context. LncRNAs are mainly classified according to the physical location of the genome. Therefore, they are dubbed as natural antisense transcripts, overlapping transcripts, and intronic or exonic transcripts, depending on the genomic arrangement of the lncRNA for their closer protein-encoding gene locus [9]. LncRNAs have emerged as essential players in the regulatory landscape of eukaryotic cells, with their diverse mechanisms of action contributing to the fine-tuning of gene expression, chromatin organization, and cellular processes [10] (Figure 1). They can evict chromatin-associated proteins, such as DNA methyltransferases, to maintain transcriptional activity at specific loci [11]. By recruiting mediator complexes to enhancers, lncRNAs stabilize chromatin looping, facilitating transcriptional activation [12]. Conversely, specific lncRNAs disrupt enhancer–promoter interactions, repressing gene expression [13,14]. Additionally, lncRNAs direct chromatin-modifying complexes to genomic targets via DNA–RNA triplex structures, enabling site-specific epigenetic regulation [15,16]. Scaffolding allows lncRNAs to coordinate multiprotein complexes, such as histone modifiers, to synchronize transcriptional repression or activation [17,18]. They also sequester proteins or miRNAs, blocking their activity or preventing miRNA-mediated mRNA degradation [10,11]. Post-transcriptionally, lncRNAs influence RNA processing by modulating alternative splicing through interactions with splicing factors [19,20]. Finally, lncRNAs stabilize mRNAs by recruiting RNA-binding proteins that protect transcripts from degradation, ensuring proper cellular function [21,22]. These multifaceted roles highlight lncRNAs as critical regulators of cellular processes (Figure 1).

In cancer cells, lncRNAs are essential regulators of genes involved in tumor development and progression. These RNA molecules regulate critical cellular functions in cancer cells including cell proliferation, apoptosis, angiogenesis, vasculogenic mimicry, therapy response, migration, invasion, and metastasis [23]. During the last decade, many lncRNAs from tumor tissues and cancer cell lines have been identified using DNA microarrays and high-throughput RNA-sequencing technologies. However, only a small percentage of lncRNAs have been functionally characterized. The increasing data regarding the genomic expression in tumors have allowed us to dissect their functions in tumor biology and decipher the impact on cancer hallmarks, making them potential biomarkers for cancer diagnosis and prognosis (Figure 2). However, the functions of lncRNAs in regulating genes involved in VM activation in different human cancers have not been fully elucidated. Several questions remain to be addressed in the field: (i) what are the action mechanisms of lncRNAs during the VM process? (ii) are their functions restricted to establish specific lncRNA/miRNA/mRNA axes? Or can they act by different processes? Furthermore, (iii) can they represent accurate biomarkers for prognosis, diagnosis, and promising therapeutic targets reaching clinical trials? In the present review, we discuss the actual knowledge about the role of lncRNAs in VM based on in vitro and in vivo evidence and delineate future directions for research on this emerging cancer hallmark.

## 2. LncRNAs Regulate Vasculogenic Mimicry in Human Cancers

VM is a unique process where tumor cells mimic endothelial cells to form vessel-like structures, providing an alternative route for blood supply and facilitating tumor growth and metastasis [2]. This process is independent of angiogenesis, which forms new blood vessels from pre-existing ones and poses a significant challenge to conventional anti-angiogenic therapies [3]. The presence of VM has been linked to increased tumor grade, invasiveness, and resistance to treatment, ultimately contributing to poor patient outcomes. In this regard, the dysregulation of the expression of lncRNAs has been implicated in various cancers, contributing to tumorigenesis, progression, and resistance to chemotherapy [23]. Understanding the molecular mechanisms underlying VM, particularly the involvement of lncRNAs, is crucial for developing novel therapeutic strategies to target this process and improve oncologic patient outcomes.

According to the published evidence, several commonalities can be delineated for the mechanisms of action of lncRNAs during the VM process: (i) they mainly act as sponges of miRNAs; (ii) they function by establishing specific lncRNA/miRNA/mRNA axes, and (iii) they frequently modulate proteins involved in ECM remodeling (e.g., metalloproteinases (MMPs)), influencing cell migration and invasion. For instance, LINC00339 promotes cell proliferation, invasion, and VM by sponging miR-539-5p, resulting in TWIST1 depression, which, in turn, activates cell migration via metalloproteinases 2 and 14 (MMP2, MMP14) transcriptional activation [24]. Table 1 summarizes the overexpressed lncRNAs with miRNA-sponging functions influencing the development of VM in diverse types of cancers.

Another subset of lncRNAs performs their functions through diverse mechanisms that do not involve miRNA sponging including molecular scaffolding, protein interactions, mRNA stabilization or degradation, and alternative splicing regulation (Table 2). Regardless of the mechanism employed, either miRNA sponging or a different mechanism, lncRNAs have emerged as key regulators of VM. Interestingly, most lncRNAs are upregulated in tumor tissues, suggesting they mainly act as oncogenes. Next, we describe lncRNA functions reported in specific tumors, focusing on their role as proliferation, invasion, and VM regulators. Here, we summarize the cancers with more than one lncRNA functionally characterized.

### 2.1. LncRNA Functions in Glioma

Glioma is the most common primary brain tumor, originating in glial cells that support and protect neurons in the brain, and accounts for approximately 81% of malignant brain tumors [47]. These are highly aggressive tumors that can be challenging to treat, with poor prognosis for patients, particularly those with high-grade gliomas. Despite advancements in treatment modalities, the average lifespan of newly diagnosed patients remains less than 2.5 years, with a 5-year survival rate below 5% [47]. Glioma is considered a highly angiogenic tumor, so anti-angiogenic therapy constitutes an important strategy to inhibit the progression of these tumors. However, anti-angiogenic therapy could cause the tumor to increase its adaptation to a hypoxic microenvironment, leading to VM development and ensuring that glioma cells obtain the nutrients and oxygen to grow.

Diverse reports highlight the role of lncRNAs in regulating VM in glioma (Figure 3). Interestingly, posttranscriptional modification of lncRNA molecules and its effects in VM have been reported. For instance, HOX antisense intergenic RNA myeloid 1 (HOTAIRM1) is an lncRNA that plays an oncogenic role in various cancers promoting VM. Research suggests that HOTAIRM1 promotes VM in glioma through its overexpression, contributes to aggressive behavior, and increases cell proliferation, migration, invasion, and the promotion of tumor growth. While the exact mechanisms by which HOTAIRM1 promotes VM are still under investigation, the evidence suggests that this lncRNA plays a crucial role in this process through METTL3 [38]. METTL3 increases HOTAIRM1 stability through N6-methyladenosine (m6A) modification, essentially attaching a chemical tag to HOTAIRM1, making it more resistant to degradation, thus increasing its lifespan. This process leads to increased IGFBP2 expression [38], which could promote the expression of proteins like CD144 and MMP2, which are involved in cell adhesion, migration, and ECM breakdown, all crucial steps in forming the channel-like structures that characterize VM [38].

#### LncRNAs Acting as ECM Remodeling Factors During VM in Glioma

Remarkably, lncRNAs function via the modulation of proteins involved in ECM remodeling including MMPs and transcription factors. For instance, LINC00339 promotes VM formation by regulating the miR-539-5p/TWIST1/MMP axis in U87 and U251 glioma cells. LINC00339 functions as a ceRNA for miR-539-5p, preventing it from binding to its target mRNAs [24]. When LINC00339 levels are low, miR-539-5p is free to bind to the mRNA of TWIST1, thus decreasing its protein levels. In contrast, LINC00339 indirectly increases the TWIST1 levels by suppressing miR-539-5p, increasing MMP2 and MMP14 expression, and activating ECM remodeling and VM formation [24].

A second lncRNA that modifies the expression for proteins involved in ECM re-modeling to stimulate VM is OIP5-AS1. A study reported that the SUMOylation of IGF2BP2 increased its stability. This modification protects IGF2BP2 from degradation, leading to its accumulation in the cell. IGF2BP2, now stabilized, interacts with and stabilizes the lncRNAOIP5-AS1, increasing its levels in glioma cells and acting as a sponge for miR-495-3p. This prevents their binding to their target mRNAs, effectively augmenting their expression [25]. When not sequestered via OIP5-AS1, miR-495-3p targets the mRNAs of HIF-1α and MMP14, two crucial proteins for VM formation [25]. HIF-1α promotes angiogenesis and VM, while MMP14 helps break down the ECM, facilitating cell migration and VM channel formation. The lncRNA, OIP5-AS1, plays a central role in this process, acting as a sponge for miR-495-3p, shifting the balance in favor of VM formation [25].

A further lncRNA that modifies the expression for proteins involved in ECM remodeling to stimulate VM is LOXL1-AS1. The interplay between LOXL1-AS1, TIAR, miR-374b-5p, and MMP14 forms a complex regulatory network that influences VM in glioma. TIAR, an RNA-binding protein, directly interacts with the lncRNA LOXL1-AS1, destabilizing it [26]. LOXL1-AS1 functions as a ceRNA, binding to and sequestering miRNAs, preventing them from binding to their target messenger RNAs (mRNAs). In this case, LOXL1-AS1 acts as a sponge for miR-374b-5p, reducing its availability to regulate its target genes. When not sequestered via LOXL1-AS1, miR-374b-5p can bind to the mRNA of MMP14, which codifies a protein involved in the breakdown of the ECM, a crucial process for VM formation, leading to mRNA degradation. The interactions between these molecules ultimately impact VM formation, since high TIAR levels lead to low LOXL1-AS1, which allows miR-374b-5p to suppress MMP14, hindering VM. Conversely, low TIAR levels allow LOXL1-AS1 to sponge miR-374b-5p, increasing MMP14 expression and promoting VM [26].

The lncRNA HULC, which has been identified as a key player in glioblastoma development, contributes to the tumor’s aggressiveness and growth and modifies the expression of proteins involved in ECM remodeling [27]. HULC stimulates the EMT process, allowing glioma cells to detach, move freely, and form channel-like structures characteristic of VM. That is important since there is a positive correlation between the number of VM structures and disease progression in glioblastoma patients. In agreement, higher HULC expression has been associated with shorter progression-free survival periods in patients [27]. This makes HULC a potential target for therapeutic interventions to disrupt VM and hinder glioblastoma progression.

Another lncRNA that regulates MMPs and ECM remodeling is LINC00707. This molecule regulates VM through an intricate process that begins with the interplay with HNRNPD, an RNA-binding protein that interacts with the mRNA of ZHX2. This interaction enhances ZHX2 mRNA stability and protein levels [17]. ZHX2, acting as a transcription factor, binds to the promoter region of LINC00707, reducing its expression. LINC00707 functions as a ceRNA for miR-651-3p that, when not bound by LINC00707, targets the mRNA of SP2, a transcription factor involved in various cellular processes including VM formation. This binding leads to the degradation of SP2 mRNA [28]. SP2 directly promotes VM formation by binding to the promoter regions of genes involved in VM such as MMP2, MMP9, and VE-cadherin. These genes contribute to the breakdown of the ECM and the formation of vessel-like structures [28]. Thus, HNRNPD indirectly suppresses LINC00707 by stabilizing ZHX2. This suppression allows miR-651-3p to inhibit SP2, a promoter of VM.

A further lncRNA that regulates the ECM remodeling process is ZRANB2. It has been reported that the ZRANB2/SNHG20/FOXK1 axis plays a crucial role in regulating VM formation in the U87 and U251 glioma cell lines. This regulatory mechanism involves the lncRNA SNHG20, a key intermediary. The RNA-binding protein ZRANB2 directly interacts with SNHG20, enhancing its stability and increasing its levels within glioma cells [36]. SNHG20 influences FOXK1 mRNA degradation through the STAU1-mediated mRNA decay pathway. FOXK1 expression negatively correlates with the pathological grade of glioma and negatively regulates the expression of VM-related molecules MMP1, MMP9, and VE-cadherin [36]. This finding highlights the potential of targeting this axis, particularly SNHG20, as a novel therapeutic strategy for treating glioma by disrupting its ability to form VM and sustain its growth.

Finally, another important lncRNA in glioma development is HCG15, which participates in the PABPC5/HCG15/ZNF331 feedback loop regulating VM by influencing key protein expression [37]. It has been reported that PABPC5, a protein involved in mRNA stability, binds to HCG15, increasing its levels within glioma cells. HCG15 influences ZNF331 mRNA degradation through the STAU1-mediated mRNA decay pathway. The transcription factor ZNF331 binds to the promoter region of the PABPC5 gene, inhibiting its transcription. This inhibition reduces the PABPC5 protein levels, completing the feedback loop. Thus, high PABPC5 and HCG15 levels, which lead to low ZNF331 levels, promote VM [37].

### 2.2. LncRNA Functions in Lung Cancer

Lung cancer remains the first cause of cancer-related mortality. It accounts for 2,480,301 new cases, resulting in 1,817,172 deaths in 2022 [48]. It is a heterogeneous and very aggressive disease. Non-small cell lung cancer (NSCLC) is the most common histological subtype (85% of cases). NSCLC is classified into three subtypes: lung adenocarcinoma, squamous cell carcinoma (LUSC), and large cell carcinoma (LCC) based on the histopathology, cells, site of origin, and molecular profile. Risk factors include smoking, air pollution, exposure to environmental hazards (radon and asbestos), and genetic predisposition, which also contribute to the disease burden [49]. Even with advances in diagnoses and new therapies, the prognosis for this cancer type remains unsatisfactory, making the identification of new biomarkers and therapeutic targets imperative.

In this context, several studies have linked lncRNAs with VM in lung cancer (Figure 3), for example, the lncRNA MALAT1 (metastasis-associated lung adenocarcinoma transcript 1), which acts as a ceRNA by sponging miR-145-5p, a microRNA with tumor-suppressive properties. This interaction suppresses miR-145-5p’s ability to target and downregulate NEDD9 mRNA, which codifies a protein involved in cell adhesion, migration, and invasion [29]. Elevated NEDD9 levels, facilitated by the MALAT1/miR-145-5p interaction, contribute to the EMT process, allowing cancer cells to acquire migratory and invasive characteristics, further promoting VM formation. ERβ can indirectly promote VM by modulating this axis. ERβ promotes higher NEDD9 levels by increasing MALAT1 expression, fostering VM formation [29].

LINC00312 plays a crucial role in inducing VM in lung adenocarcinoma by directly interacting with YBX1, a transcription factor involved in various cellular processes including angiogenesis and cell proliferation [39]. The direct interaction between LINC00312 and YBX1 ultimately leads to increased migration, invasion, and VM formation in lung adenocarcinoma A549 cells [39]. On the other hand, LINC01555 promotes chemoresistance in SCLC by modulating the miR-122-5p/CLIC1/Amot-p130 axis, ultimately influencing VM formation [30]. LINC01555 acts as a ceRNA for miR-122-5p, effectively sponging it and preventing its interaction with its target mRNA, CLIC1. Consequently, LINC01555 upregulation leads to increased CLIC1 expression. CLIC1, in turn, negatively regulates Angiomotin-p130. Therefore, increased CLIC1 levels lead to decreased Amot-p130 expression and activity. Amot-p130, a known suppressor of VM, plays a crucial role in chemotherapy sensitivity [30]. Its downregulation, mediated by the LINC01555/miR-122-5p/CLIC1 axis, promotes VM formation. This process contributes to chemoresistance by providing SCLC cells with an alternative nutrient and oxygen supply pathway, enabling them to evade the cytotoxic effects of chemotherapy drugs [30].

Another lncRNA reported is LINC00987, which is frequently downregulated in lung adenocarcinoma due to promoter hypermethylation, which correlates with poor prognosis. Mechanistically, LINC00987 directly binds to SND1, promoting its phosphorylation and subsequent degradation [35]. This interaction inhibits lung adenocarcinoma cell proliferation, migration, invasion, and VM formation.

### 2.3. LncRNA Functions in Gastric Cancer

Gastric cancer is the fifth most common cancer worldwide in terms of incidence and mortality, with an estimated 968,350 new cases and 659,853 deaths in 2022 [48]. Gastric cancer can be divided into two distinct entities: cardia (upper stomach) and non-cardia (lower stomach). H. pylori infection is associated with non-cardia gastric cancer. However, only a proportion of these patients will develop cancer, which is explained by differences in bacterial genetics, host genetics, age of acquisition of infection, and environmental factors including the consumption of alcohol, tobacco, and some foods [48].

Several lncRNAs have been implicated in VM formation in gastric cancer (Figure 3). Among these, metastasis-associated lung adenocarcinoma transcript 1 (MALAT1), also known as nuclear-enriched abundant transcript 2, promotes angiogenesis and VM formation in patients with gastric cancer [40]. In agreement, MALAT1 knockdown in BGC823 and SGC901 gastric cancer cells reduces migration, invasion, VM formation, and angiogenesis. MALAT1 promotes VM formation and angiogenesis via the VE-cadherin/β-catenin complex, ERK/MMP, and FAK/paxillin signaling. The mechanism involves the activation of VM and angiogenesis markers, such as β-catenin, VE-cadherin, MMP2, and MMP9, and the phosphorylation of signaling proteins p-ERK, p-paxillin, and p-FAK [40].

Plasmacytoma variant translocation 1 (PVT1) is another lncRNA involved in VM formation in gastric cancer. PVT1 overexpression promotes VM formation and high VE-cadherin expression in vivo. EMT is regulated by transcription factors including SLUG [28]. In addition, PVT1 overexpression increased the expression levels of EMT- and VM-related genes such as Slug, VE-cadherin, N-cadherin, Vimentin, and MMP2 while decreasing that of E-cadherin. PVT1 interacts with and recruits STAT3 to the SLUG promoter, increasing the transcriptional activity of Slug to promote EMT and VM formation. In turn, STAT3 can also induce PVT1 expression, thus creating a positive feedback loop. In summary, PVT1 plays a pivotal role by promoting EMT and VM formation in gastric cancer through the PVT1/STAT3/Slug axis [41].

RPMS1 is an EBV-induced lncRNA highly overexpressed in gastric cancer cell lines. CXCL8 is also overexpressed in this cell model, activating the NFKβ signaling cascade to induce VM formation. RPMS1 silencing reduces VM formation, proliferation, and migration in EBV-infected cells, which can be reversed by CXCL8 treatment [42]. RPMS1 increases CXCL8 expression to induce VM formation through the NFK-β signaling pathway. NFK-β signaling promotes CCL8-induced VM formation through the overexpression of MMP1, MMP9, and TWIST, and NFκ-β signaling pathway inhibitors repress this effect. RPMS1 knockdown decreases H3k27me mark expression, indicating that RPMS1 may regulate CXCL8 expression through an epigenetic mechanism [42].

The oncogenic function of lncRNA UCA1 has been widely documented in GC, involving the activation of signaling pathways such as PI3K/AKT and AKT/GSK3-β, which are involved in proliferation and metastasis. Furthermore, UCA1 may also act as an miRNA sponge by regulating downstream target proteins. In this context, a study by Lu Y et al. (2024) revealed high expression levels of UCA1 in gastric cancer and its association with a poor prognosis [31]. Furthermore, high UCA1 expression levels are associated with tumor size, invasion depth, metastasis to lymphoid nodules, and TNM stage. UCA1 acts as a sponge for miR-1-3p, an miRNA with a tumor suppressor function that significantly inhibits VM growth and formation. Ectopic miR-1-3p expression produces apoptosis in SGC7901 and AGS gastric cancer cells by increasing the BAX, BAD, and P53 protein expression levels. In addition, UCA1 induces VM formation, while miR-1-3p overexpression induces the reverse effect [31]. These findings highlight the role of UCA1 and its downstream target, miR-1-3p, in tumor growth and progression.

### 2.4. LncRNA Functions in Breast Cancer

Breast cancer is the most frequently diagnosed cancer and is the leading cause of death in women, with an estimated 2.3 million new cases and 66,000 deaths according to figures provided by Globocan 2022 [48]. Breast cancer is a heterogeneous disease and is usually considered as a group of diseases with a distinct genetic profile and clinical behavior. Histologically, breast cancer is divided into two main subtypes: invasive ductal carcinoma (the most common) and invasive lobular carcinoma. Immunohistochemical classification divides breast cancers into five molecular subtypes according to the expression of their hormonal receptors: estrogen receptor (ER) and progesterone receptor (PR), human epidermal growth factor receptor (HER2), and Ki-67. Thus, the subtypes are luminal A (ER+, PR+, HER-2 negative, Ki-67 < 14%); luminal B HER2- (ER+, PR+, HER-2 negative, Ki-67 ≥ 14%); luminal B HER2+ (ER+, PR+, HER-2+, any Ki-67); HER2 enriched (ER-, PR-, HER2+++); and triple negative (TNBC), (ER-, PR-, HER2-) breast cancer [50]. Of these, TNBC is the most aggressive and has the worst prognosis with no therapeutic targets. Risk factors for breast cancer include both exogenous and endogenous factors including age, personal and family history, use of hormonal therapy, early menarche, late menopause, breast density, and genetic risk factors such as the presence of mutations [48].

Two recent studies have reported VM-associated lncRNAs in breast cancer (Figure 4). For example, HOTAIR was found to be highly expressed in breast cancer tissues and showed a significant increase in cancer cell lines under hypoxic conditions [32]. Hypoxia is a process that induces VM formation. HOTAIR silencing in MDA-MB-231 and Hs-578t cancer cell lines inhibits 3D channel-like network formation and cell migration. HOTAIR functions as a ceRNA that sequesters miR-204, leading to FAK derepression, a protein tyrosine kinase involved in cell migration, resulting in VM activation [32].

Another lncRNA implicated in VM formation in TNBC is the TP73 antisense RNA 1 (TP73-AS1). TP73-AS1 is highly expressed in VM-positive triple TNBC cells [33]. TP73-AS1 knockdown inhibits VM formation in the MDA-MB-231 cell line. TP73-AS1 acts as a sponge for miR-490-3p, preventing binding to its target genes. In addition, miR-490-3p overexpression in the MDA-MB-231 cell line suppresses VM formation by targeting transcription factor TWIST1. In conclusion, TP73-AS1 induces VM formation by releasing TWIST from miR-490-3p-induced transcriptional repression.

### 2.5. LncRNA Functions in Renal Cell Carcinoma

Renal cell carcinoma (RCC) has slowly increased in recent decades, accounting for 90% of all renal cancers, with 5-year survival rates of around 75% [51]. Tobacco use and obesity are the main associated risk factors. RCC represents a heterogeneous disease that can be subclassified considering the histopathological and molecular characteristics. Clear-cell RCC represents the most prevalent subtype, with 70–80% of cases characterized by its high aggressiveness and low survival rates [51].

Many lncRNAs have critical roles in RCC therapy response (Figure 4); for example, treatment with sunitinib, an angiogenesis inhibitor, induces VM formation both in vivo and in vitro. The mechanism activates the lncRNA-ECVSR/ERβ/Hif2-α axis [43]. Sunitinib treatment induces ECVSR, which in turn increases ERβ mRNA stability by directly binding to its 3′UTR region. Interestingly, ECVSR contains estrogen response elements (EREs) in its promoter region, allowing for regulation via ERβ, thus creating a positive feedback loop. Erβ induces VM formation, at least partially via the induction of the hypoxia-inducible transcription factor, Hif2-α, which regulates the expression of CSC markers such as SOX2, Oct4, and Nanog in A498 cells. Using an ERβ-specific inhibitor via shRNA or an Erβ antagonist combined with sunitinib treatment led to the inhibition of VM formation in an in vivo RCC model [43].

SERB is another lncRNA implicated in regulating ERβ via binding to its promoter region. SERB overexpression has a poor prognosis in RCC patients [44]. Moreover, functional studies have demonstrated that the ectopic expression of SERB increases VM formation in the RCC cell line A498. The binding of SERB to ERβ promotes ERβ expression, and ERβ knockdown reverses VM formation induced by SERB. ZEB1 possesses EREs in its promoter region, allowing its messenger level regulation via ERβ [44]. In summary, SERB acts as an oncogenic lncRNA inducing VM formation in RCC through the SERB/ERβ/ZEB1 axis, which could be helpful in developing a new therapy in RCC.

Angiogenesis and VM are processes closely associated with metastatic progression, contributing to increased aggressiveness and poor prognosis. Androgen receptor (AR) inhibitors may improve the response to sunitinib [45]. Treating human RCC cell lines 7860 and SW839 with an AR agonist and antagonist increased and suppressed VM formation, respectively. A recent study reported that AR induced VM formation in RCC through a mechanism involving the transcriptional regulation of the lncRNA TANAR, which has androgen response elements in its promoter, allowing AR to increase its expression directly at the transcriptional level. TANAR overexpression increases VM formation, even after AR knockdown, while its silencing inhibits VM formation. RA-induced VM activation indirectly involves TWIST1, a gene that promotes the EMT process, and VM. TANAR regulates TWIST1 expression, inhibiting the UPF1–TWIST1 mRNA interaction, which prevents the nonsense-mediated decay of TWIST1 mRNA, preventing its degradation through directly binding to the 5′UTR of TWIST1. Furthermore, TWIST1 transcriptionally regulates VE-cadherin expression [45].

### 2.6. LncRNA Functions in Colorectal Cancer

Colorectal cancer (CRC) is the third most prevalent type of cancer as well as the second leading cause of cancer-related death worldwide, which accounts for 903,859 deaths and 1,926,118 new cases [48]. This cancer originates in the colon or rectum. It begins with the development of adenomatous polyps that grow slowly, giving a window of opportunity to detect this cancer at an early stage. The histological classification of CRC includes three main subtypes: adenocarcinoma, mucinous adenocarcinoma, and signet ring cell carcinoma. The first is the most common. CRC is a complex and genetically heterogeneous disease that drastically influences the patient’s treatment [52]. Several recent investigations have shown that lncRNAs play an essential role in CRC initiation, progression, and metastasis (Figure 4) [34,46].

An unexpected functional link between VM and glucose metabolism has been described in CRC. Glycolysis plays a pivotal role in SOX2-induced VM induction. SOX2 binds to the lncRNA AC005392.2 promoter and reduces the H3K27me3 transcriptional repression epigenetic mark, activating AC005392.2 expression in CRC cells [46]. AC005392.2 overexpression increases the stability of the glucose transporter GLUT1 by increasing its SUMOylation, leading to decreased ubiquitination and preventing proteasome-mediated degradation. GLUT1 blockade inhibits the expression of VM-associated molecules VE-cadherin and EPH2A. AC005392.2, GLUT1, and EPH2A expression levels were associated with poor prognosis [46]. Therefore, blocking glycolysis could be an effective therapeutic strategy to inhibit VM formation in CRC.

The lncRNA NORAD plays a crucial role in CRC progression and chemoresistance by modulating processes such as VM and EMT. Exposure to hypoxic conditions in CRC cell lines HCT116 and SW480 increases the formation of tube-like structures as well as resistance to 5-FU while increasing NORAD expression and VM development [34]. The suppression of NORAD expression leads to increased apoptosis and E-cadherin expression levels while decreasing the HIF-1α expression levels. This transcription factor regulates gene expression in cellular adaptation to hypoxia. NORAD functions as a sponge for miR-495-3p, preventing binding to its target mRNA HIF-1α. In summary, the NORAD/miR-495-3p/HIF-1α axis plays a fundamental role in VM induction in CRC, so NORAD could function as a new therapeutic target in CRC [34].

## 3. LncRNAs as Therapeutic Targets

Preclinical models for the study of VM, mainly based on xenograft mouse models, have demonstrated the potential of lncRNAs as therapeutic targets for cancer treatment. Most of the lncRNAs discussed above have been targeted for in vivo assays in mouse models. In vivo studies have shown that lncRNA knockdown significantly inhibits tumor growth and VM formation. These exciting findings suggest that lncRNA depletion alone, or in combination with FDA-approved drugs, could be a proper therapeutic strategy for inhibiting tumor growth, particularly in chemoresistant tumors where antiangiogenic therapies have failed due to VM development. Several studies have exemplified these possibilities; for instance, downregulated linc01555 restrained tumor growth and cisplatin resistance in vivo [30]. Another example is the lncRNA-ECVSR overexpressed in RCC; where the treatment with sunitinib increased ECVSR-mediated Erβ mRNA stability, leading to overexpressed CSC markers. Pre-clinical studies using RCC mouse xenografts showed that combining sunitinib with the anti-estrogen PHTPP could increase sunitinib’s efficacy with reduced VM formation [43]. These studies highlight the potential therapeutic value of targeting specific lncRNAs to inhibit VM formation in cancer.

Antisense oligonucleotides (ASOs) are a promising therapeutic strategy to target and inhibit specific RNAs in cancer cells, resulting in premature transcriptional termination, degradation, decreased proliferation, and chemoresistance inhibition [53]. Another tool that could be used to target specific RNAs and inhibit VM formation is CRISPR-CasRx gene editing technology. For example, in bladder cancer, CRISPR-CasRx-mediated knockdown of LINC00341 increased the expression of Bax, p21, and E-cadherin, leading to decreased proliferation and increased apoptosis [54]. However, it is essential to consider some limiting factors in using these tools, such as degradation via RNases, tissue-specific targeting, and hepatotoxicity observed using certain ASOs [53,54,55]. Further research is needed to evaluate the effectiveness of these tools in inhibiting VM in vivo and to develop more effective therapeutic strategies.

## 4. Conclusions and Future Perspectives

Cancer is considered a complex genetic and epigenetic disease. However, a new exciting theory proposes that it is an ecological disease: a multidimensional spatiotemporal “unity of ecology and evolution” pathological ecosystem [56]. VM is formed by adopting multiple cellular phenotypes, including channel-like 3D structures or endothelial-like characteristics, reflecting an adaptative behavior that resembles an ecotype [56]. The mechanisms governing VM have not been well-described. Therefore, this review aimed to enhance our understanding of the molecular mechanisms of lncRNAs in VM formation in cancer. Recent data highlight the essential role of these non-coding RNAs in modulating genes that activate VM in diverse types of neoplasia. LncRNAs not only modulate VM, but are also key molecules in the development of cancer hallmarks and pathogenesis. Future studies should be used to identify lncRNA signatures from cancer tissues positive for VM, which will allow us to understand the tumor biology behind the VM mechanisms. It will also be crucial to perform multi-institutional studies, with large patient cohorts, including independent data validation between different research institutions and laboratories, to define the potential of lncRNAs as novel molecular biomarkers of progression and prognosis. Finally, a more exhaustive functional validation of lncRNAs as oncogenes or tumor suppressors should be prioritized to foster the use of lncRNAs as effective therapeutic targets impacting VM and cancer hallmarks.

## Figures and Tables

**Figure 1 cells-14-00616-f001:**
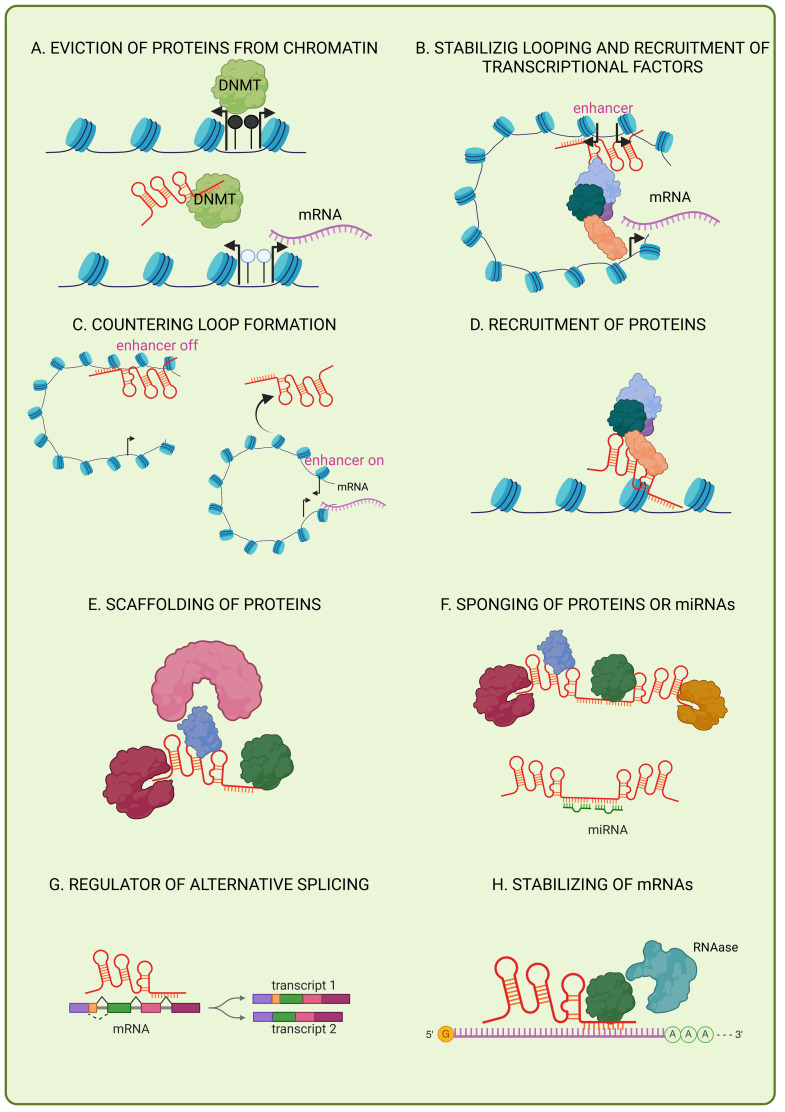
Mechanisms of action of lncRNAs. (**A**) LncRNAs can evict proteins from chromatin and inhibit DNMT activity at promoter regions, blocking DNA methylation to enable transcriptional activation. (**B**) LncRNAs facilitate enhancer-mediated transcription by recruiting the mediator complex, stabilizing chromatin looping for gene expression. (**C**) Enhancer-transcribed lncRNAs may disrupt spatial enhancer–promoter interactions, resulting in the transcriptional suppression of adjacent genes. (**D**) LncRNAs can recruit proteins to induce cellular processes. (**E**) LncRNAs act as modular scaffolds to tether functionally related proteins, enabling cooperative regulatory activities. (**F**) LncRNAs can also sequester proteins or miRNAs to block their regulatory functions. (**G**) LncRNAs can modulate pre-mRNA processing by binding nascent transcripts to alter splice site selection. (**H**) LncRNAs enhance mRNA stability through interactions with RNA-binding proteins, which shield transcripts from degradation. Figure created in BioRender (https://www.biorender.com).

**Figure 2 cells-14-00616-f002:**
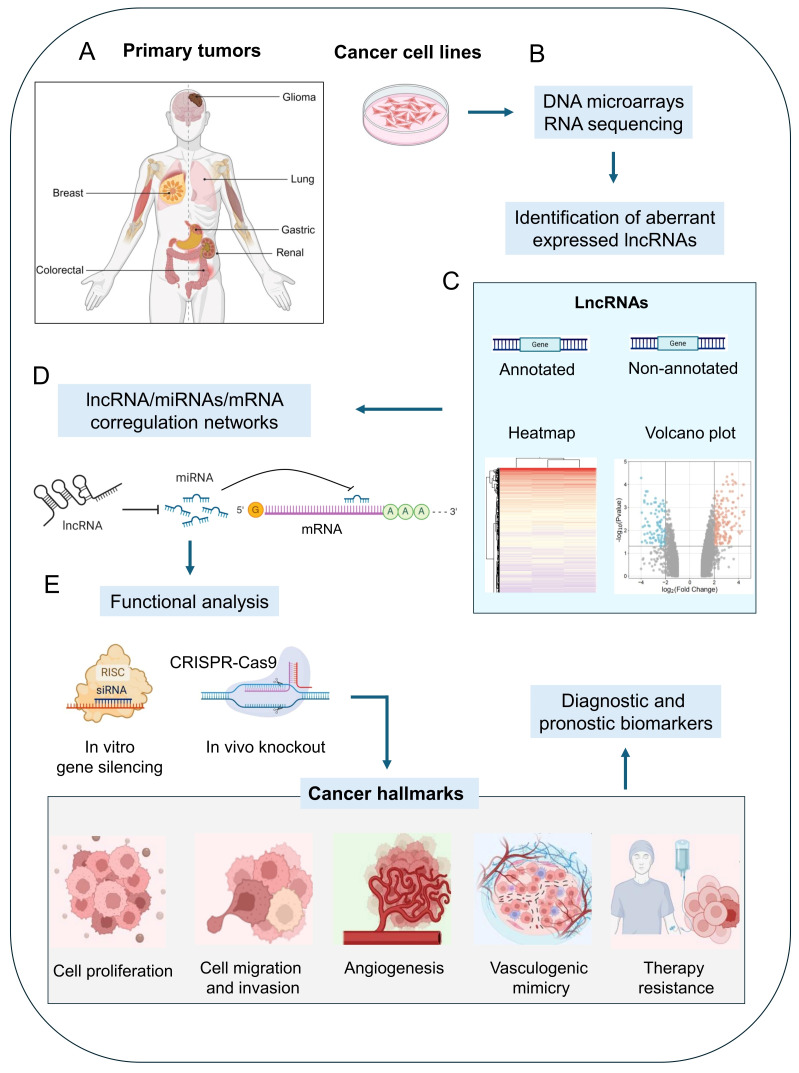
A typical workflow for lncRNA characterization in cancer. (**A**) The studies initiate with the selection of normal and tumor clinical samples from small patient cohorts, which may include primary or metastatic tumors. (**B**) Non-coding and protein-coding RNA expression profiling using DNA microarrays or RNA-sequencing technologies allows for the identification of the abundance of annotated or novel non-annotated lncRNA transcripts in the samples. After a comparison between normal and tumor samples, the deregulated lncRNAs can be identified. (**C**) Bioinformatics analysis can predict the lncRNA/miRNA/mRNA coregulation networks and cellular processes where the RNA molecules can function. (**D**) Based on the information obtained from genomics data integration and bioinformatic analysis, an lncRNA/miRNA/mRNA axis can be selected for deep functional analysis. (**E**) Technologies for in vitro functional analysis include using siRNAs or precursors to repress or overexpress a particular lncRNA and evaluating their effects on cancer hallmarks. Functional in vivo screening using advanced CRISPR-Cas9 gene knockdown can also be implemented for both in vitro and in vivo analysis.

**Figure 3 cells-14-00616-f003:**
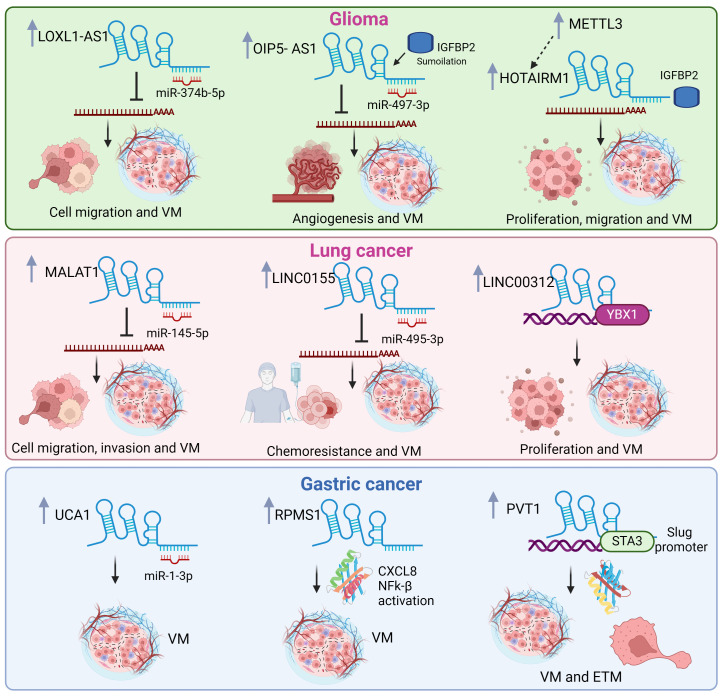
LncRNAs overexpressed in glioma, lung, and gastric cancers associated with vasculogenic mimicry. The lncRNAs and sponging microRNAs as well as the mRNAs targets of microRNAs are indicated. The cancer hallmarks regulated by the lncRNA/microRNA/mRNA axes are also denoted. Up gray arrows denote upregulation. Created in BioRender (https://www.biorender.com).

**Figure 4 cells-14-00616-f004:**
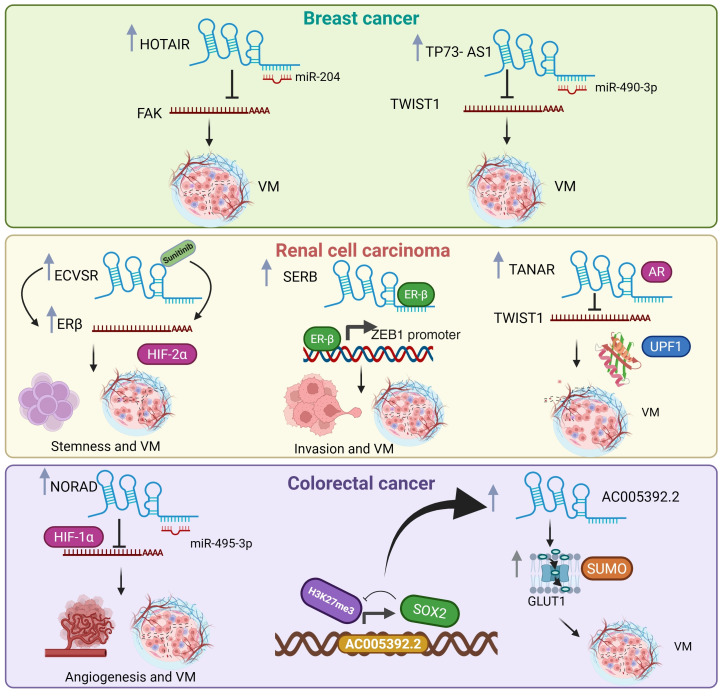
LncRNAs associated with vasculogenic mimicry in breast, colorectal, and renal carcinoma. The lncRNAs and sponging microRNAs as well as the mRNAs targets of microRNAs are indicated. The cancer hallmarks regulated by the lncRNA/microRNA/mRNA axes are denoted. Up gray arrows denote upregulation. Figure created in BioRender (https://www.biorender.com).

**Table 1 cells-14-00616-t001:** Upregulated lncRNAs with sponge functions promoting vasculogenic mimicry.

lncRNA	Cancer	Target	Mechanisms of Action	References
LINC00339	Glioma	miR-539-5p	Promotes proliferation, invasion, and VM by sponging miR-539-5p and increasing TWIST1 expression	[24]
OIP5-AS1	Glioma	miR-495-3p	Promotes angiogenesis, migration, and VM acting as a sponge for miR-495-3p, regulating HIF-1α and MMP14	[25]
LOXL1-AS1	Glioma	miR-374b-5p	Involved in ECM degradation; downregulates miR-374b-5p and increases MMP14 expression	[26]
HULC	Glioma	Unknown	Stimulates the EMT and tube formation	[27]
LINC00707	Glioma	miR-651-3p	Induces ECM degradation and VM formation, reducing the expression of miR-651-3p and increasing its target gene SP2	[28]
MALAT1	Lung	miR-145-5p	Stimulates cell invasion and VM formation and acts as a competitive endogenous RNA (ceRNA) by sponging miR-145-5p	[29]
LINC01555	Lung	miR-122-5p	Promotes angiogenesis and the chemoresistance miR-122-5p/CLIC1 axis in vivo	[30]
UCA1	Gastric	miR-1-3p	Induces proliferation and metastasis, sponges miR-1-3p, and activates PI3K/AKT and AKT/GSK3-β signaling	[31]
HOTAIR	Breast	miR-204	Promotes cell migration, is highly expressed under hypoxic conditions, and adsorbs miR-204; FAK is the target gene of miR-204	[32]
TP73-AS1	Breast	miR-490-3p	Involved in VM formation; acts as an miR-490-3p sponge by quenching TWIST1	[33]
NORAD	Colorectal	miR-495-3p	Modulates VM and EMT, acting as a sponge for miR-495-3p, preventing binding to its target mRNA HIF-1α	[34]

VM, vasculogenic mimicry; ECM, extracellular matrix; EMT, epithelial–mesenchymal transition; ceRNA, competitive endogenous RNA.

**Table 2 cells-14-00616-t002:** VM-associated lncRNAs have different functions than sponges.

LncRNA	Cancer	Validated Target	Function in VM	Mechanisms of Action	References
**Downregulated**					
LINC00987	Lung	SND1	Inhibitor	Inhibits cell proliferation, migration, and invasion, promoting SND1 phosphorylation and degradation	[35]
**Upregulated**					
HCG15	Glioma	ZNF331	Promoter	Induces proliferation, migration, invasion, and VM formation and influences the degradation of ZNF331 mRNA, a PALB transcription inhibitor	[36]
SNHG20	Glioma	FOXK1	Promoter	Promotes proliferation, migration, invasion, VM, FOXK1 mRNA degradation, downregulating the MMP1, MMP9, and VE-cadherin levels	[37]
HOTAIRM1	Glioma	IGFBP2	Promoter	Induces proliferation, migration, invasion, and VM by regulating IGFBP2 expression	[38]
LINC00312	Lung	YBX1	Promoter	Promotes migration and invasion by directly binding to the YBX1 protein	[39]
MALAT1	Gastric	Unknown	Promoter	Increases vascular permeability and angiogenesis by activating the VE-cadherin/β-catenin complex, ERK/MMP, and FAK/paxillin signaling	[40]
PVT1	Gastric	STAT3	Promoter	Promotes EMT and VM formation by interacting and recruiting STAT3 to the SLUG promoter to induce ETM gene expression	[41]
RPMS1	Gastric	CXCL8	Promoter	Increases CXCL8 expression by interfering with H3K27me at the promoter region to induce proliferation, migration, and VM formation	[42]
ECVSR	Renal cell carcinoma	ERβ	Promoter	Sunitinib binds to and upregulates ECVSR, increasing ERβ mRNA stability and promoting a cancer stem-like cell (CSC) phenotype	[43]
SERB	Renal cell carcinoma	ERβ	Promoter	Upregulates invasion and VM formation by binding to the Erβ promoter region, increasing its expression. ZEB1 is the transcriptional target of ERβ	[44]
TANAR	Renal cell carcinoma	Twist	Promoter	The androgen receptor (AR) upregulates TANAR, which in turn inhibits TWIST1 degradation by directly binding to its 5′UTR, thereby promoting E-cadherin expression to induce VM	[45]
AC005392.2	Colorectal	GLUT1	Promoter	Activates glycolysis by binding to GLUT1 and promoting its stability	[46]

VM, vasculogenic mimicry; EMT, epithelial–mesenchymal transition; CSC, cancer stem-like cell; AR, androgen receptor.

## Data Availability

No new data were created or analyzed in this study.

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
