# Peer review of "LncRNAs Regulate Vasculogenic Mimicry in Human Cancers"

_cells, 2025, doi:10.3390/cells14080616_

Round 1

Reviewer 1 Report

Comments and Suggestions for Authors

The manuscript describes an important phenomenon, VM, which is important as an alternate way of enabling tutor perfusion. VM is also linked to treatment resistance and poor prognosis, making its understanding important for the effective treatment of cancers associated with this phenomenon.

The authors should address some major issues. The review is a summary of the literature, and does not provide critical insight. The authors should restrict the review to a group of related cancers, as the paper is too general and falls short in terms of reviewing the mechanism and the role of LncRNAs. For example, it is not logical how glioma is attributed the propensity to form VM by virtue of being a highly vascularised tumor. The mechanism needs to be clear and the argument be presented in a logical manner. Furthermore, the manuscript is disjointed.

Major language editing is required.

Comments on the Quality of English Language

There are too many grammatical errors, extensive language editing is recommended. 

Author Response

The manuscript describes an important phenomenon, VM, which is important as an alternate way of enabling tutor perfusion. VM is also linked to treatment resistance and poor prognosis, making its understanding important for the effective treatment of cancers associated with this phenomenon.

The authors should address some major issues.

  1. The review is a summary of the literature, and does not provide critical insight.

Reply: We would like to express our gratitude for your time and attention in reviewing the manuscript. The corrections and new sections have been highlighted in yellow font in the revised manuscript for ease of identification. Based on your and the other reviewers' concerns, we have revised and corrected the manuscript, improved the narrative, provided better criticisms of the actual state of the art, and pointed out the gaps in knowledge and future research directions on vasculogenic mimicry (VM) in cancer. Also, we examined the molecular mechanisms of lncRNAs (Page 2, lanes 73-89) and highlighted several commonalities in the cellular functions associated with VM between diverse cancer types (Page 5, lanes 150-159). Future directions for research focused on deciphering their function in VM are delineated. Finally, the potential of selected lncRNAs as novel therapeutic targets in RNA-based molecular interventions is also discussed (Page 16. Section Conclusions and future perspectives).

  1. The authors should restrict the review to a group of related cancers, as the paper is too general and falls short in terms of reviewing the mechanism and the role of LncRNAs. For example, it is not logical how glioma is attributed the propensity to form VM by virtue of being a highly vascularised tumor.

Reply: Thank you very much for pointing out this critical issue. As the reviewer suggested, we have shortened the review and removed two types of tumors, of which only one lncRNA has been reported to be associated with VM. We could not restrict the study to a group of related cancers, as no clear relationship existed between the reviewed tumors (e.g., no reports on VM exist on the group of gynecological cancers, except for two studies for breast cancer). Instead, we have grouped several lncRNAs with common action mechanisms, e.g., EMT and ECM remodeling via MMPs activation leading to enhanced cell migration and VM.

We apologize for the mistake in the comment about the relationship between vascularization and VM in glioma, which has been deleted because it is erroneous.

  1. The mechanism needs to be clear and the argument be presented in a logical manner. Furthermore, the manuscript is disjointed.

Reply: We thank the reviewer for the critical comments. In the revised version of the manuscript, we have better articulated the narrative and added conceptual bridges between the different paragraphs, which make the message more comprehensible to readers. Also, we have added two new informative figures, 1 and 2, about the mechanisms of action of lncRNAs (Page 3) and a typical workflow for lncRNA characterization in cancer (Page 4) to provide the readers with more context about the role and study of lncRNAs in cancer. In addition, the original figures 1 and 2 (now figures 3 and 4) were improved for clarity.

Major language editing is required.

Reply: As the reviewer suggested, the manuscript's English grammar has been reviewed and corrected using the MDPI language services. The invoice is attached as supplementary data.

Reviewer 2 Report

Comments and Suggestions for Authors

The authors comprehensively reviewed the existing literature on the role of long non-coding RNAs (lncRNAs) in regulating vasculogenic mimicry (VM) in human cancers, with particular emphasis on their potential as novel therapeutic targets for RNA-based molecular interventions.Several pivotal considerations merit attention to elevate the scholarly excellence of this manuscript.

  • Given the emphasis on the potential of lncRNAs as novel therapeutic targets for RNA-based molecular interventions, could the authors provide detailed information regarding this aspect? Are there any specific applications or clinical trials that have been conducted?
  • A significant concern is the inappropriate use of references. For instance, the patients described in Reference 27 (lines 271-274) do not correspond to gastric cancer cases as discussed.Alternatively, in line 279, the statement “has already been reported in gastric cancer [29, 30]” is inaccurate, as the patients in Reference 29 pertain to esophageal cancer, not gastric cancer.It is required to meticulously examine all the references cited in this paper to prevent misattribution.
  • Since the title “LncRNAs regulate vasculogenic mimicry in human cancers”,It is suggested that the introduction begins with an explanation of what human cancer is. Generally, cancer is regarded as a genetic disease; however, recently some researchers have proposed that “cancer as multidimensional spatiotemporal "unity of ecology and evolution" pathological ecosystem”(https://pubmed.ncbi.nlm.nih.gov/37056571/). It might provide a novel concept for our understanding of cancer process. This paper also briefly addresses VM, offering novel insights into its formation.
  • The authors elucidate the intricate roles of various LncRNAs across a spectrum of malignant tumors, particularly highlighting their influence on VM formation. I am especially intrigued to understand the precise biological behaviors and transformative changes that tumors undergo in the aftermath of enhanced VM formation in various cancers.
  • Does there exist any identical LncRNA that exerts distinct functions  including regulating VM in different tumor tissues?

Author Response

The authors comprehensively reviewed the existing literature on the role of long non-coding RNAs (lncRNAs) in regulating vasculogenic mimicry (VM) in human cancers, with particular emphasis on their potential as novel therapeutic targets for RNA-based molecular interventions. Several pivotal considerations merit attention to elevate the scholarly excellence of this manuscript.

  1. Given the emphasis on the potential of lncRNAs as novel therapeutic targets for RNA-based molecular interventions, could the authors provide detailed information regarding this aspect? Are there any specific applications or clinical trials that have been conducted?

Reply: We recognize the value of your feedback, both in shaping the manuscript and informing our future research. The corrections and new sections have been highlighted in yellow font in the revised manuscript for ease of identification.

Your question is very interesting. We have reviewed the Clinical Trials database (https://clinicaltrials.gov/), which contains the trials completed, terminated, and under development for different interventions/treatments in all types of cancer. We don’t find clinical trials for treating tumors using lncRNAs mimics/inhibitors or antisense oligonucleotide therapy targeting VM. We have highlighted these limitations as future directions in the research and potential therapeutic application of lncRNAs in cancer (Page 15, section 3. LncRNAs as therapeutic targets, Page 16, section 4. Conclusions and future perspectives).

  1. A significant concern is the inappropriate use of references. For instance, the patients described in Reference 27 (lines 271-274) do not correspond to gastric cancer cases as discussed. Alternatively, in line 279, the statement “has already been reported in gastric cancer [29, 30]” is inaccurate, as the patients in Reference 29 pertain to esophageal cancer, not gastric cancer. It is required to meticulously examine all the references cited in this paper to prevent misattribution.

Reply: We apologize for the inaccuracy in the references. We have carefully reviewed the references throughout the text and corrected the wrong ones.

  1. Since the title “LncRNAs regulate vasculogenic mimicry in human cancers”, It is suggested that the introduction begins with an explanation of what human cancer is. Generally, cancer is regarded as a genetic disease; however, recently some researchers have proposed that “cancer as multidimensional spatiotemporal "unity of ecology and evolution" pathological ecosystem ”https://pubmed.ncbi.nlm.nih.gov/37056571/. It might provide a novel concept for our understanding of cancer process. This paper also briefly addresses VM, offering novel insights into its formation.

Reply: Thank you very much for your wise observation. This novel definition of cancer is exciting and deserves discussion by the scientific community. We have reviewed the publication where the concept was formulated and observed that it was explicitly focused on nasopharyngeal carcinoma. However, as the idea is novel, we have added it to the conclusion and future perspectives section (Page 16, lane 526).

Reviewer 3 Report

Comments and Suggestions for Authors

Major revision:

The topic brought up by the authors is proven of interest but the manuscript could use more insightful synopsis and overview. It lacks a cohesive and in-depth narrative to be more informative for the reader. What are the main questions to be addressed in the field? Often the notions presented here are very vague and generic. Then authors go on about enumerating a list of lncRNAs and mirRNAs per cancer type with a lot of repetitive wording that sounds quite empty at times.

Some added value is needed compared to the current reviews already published in the topic.

Some suggestions:

The manuscript can be improved by adding comprehensive synopsis tables grouped by promoters and inhibitors of VM, adding information on the mechanism, etc, informed by a more expansive and integrative processing of the data retrieved in bibliography.

Identify and describe the main functions in the introduction before going in detail in each group (so far the main function described is as "miRNA sponges").

Extract the commonalities and the cancer-specific trends.

Which LncRNA are good candidates for RNA therapy? I was expecting a paragraph or table on that note mentioned initially.

For clarity, I suggest  present the figure's message with larger cartoons. The figures are very packed and the icons are too small.

Work on more meaningful conclusions and mention briefly some future directions and open questions.

Minor revision:

Material and methods section seems unnecessary for describing a bibliography search, which is obvious in the context of a review paper.

Fix the typos in the figure number (all pointing to figure 1).

Paragraph 3.5 on RCC: correct the typos "CRC" written many times.

Do not abuse the statement "is poorly understood/characterized" (found 3 times in a single page).

Author Response

We would like to express our gratitude for your time and attention in reviewing the manuscript. The corrections and new sections have been highlighted in yellow font in the revised manuscript for ease of identification.

  1. Does there exist any identical LncRNA that exerts distinct functions including regulating VM in different tumor tissues?

Reply: Thank you very much for pointing out this critical issue. The question is interesting. You are correct that lncRNAs could exhibit different functions in diverse tissues and have specific roles in various biological processes. This allows for a wide range of functions and roles in cellular processes. There are many lncRNAs with different functions in diverse types of cancer. This could be explained by the heterogeneous genetic background and differential expression profiles of molecules interacting (microRNAs, mRNAs, and proteins) with the lncRNAs in the different tumors.

For instance, in our study, MALAT1 was the only lncRNA that showed differential expression in 2 types of cancers: lung and gastric. According to the literature, it acts as an oncogene, so a global upregulation has been reported in various types of cancer associated with aggressive tumor behavior and poor prognosis in patients. Its biological functions are multifaceted, covering cell proliferation, angiogenesis, EMT, and drug resistance processes. In lung cancer, MALAT1 acts as a ceRNA that binds to miR-145-5p, leading to derepression of NEDD9, stimulating VM formation and cell invasion. ERβ increases MALAT1 expression by directly binding to EREs in the MALAT1 promoter. This suggests that estrogens/ERs are associated with the activation of the ERβ/MALAT1/miR145-5p/NEDD9 signaling pathway and with the progression of lung cancer.

On the other hand, in gastric cancer, increased MALAT1 expression stimulates tumor development by increasing vascular permeability and angiogenesis through the activation of the VE-cadherin/β-catenin complex, ERK/MMP, and FAK/paxillin signaling. The precise mechanisms of MALAT1 involvement in gastric cancer are not yet fully understood; further studies are required to explain its function.

Major revision:

The topic brought up by the authors is proven of interest but the manuscript could use more insightful synopsis and overview. It lacks a cohesive and in-depth narrative to be more informative for the reader.

Reply: We thank the reviewer for the critical comments. We have carefully considered the comments and have made the necessary corrections, which we hope will meet with your approval. In the revised version of the manuscript, we have better articulated the narrative and added conceptual bridges between the different paragraphs, which make the message more comprehensible to readers. Also, we have added two new figures, 1 and 2, about the mechanisms of action of lncRNAs (Page 3) and a typical workflow for lncRNA characterization in cancer (Page 4) to provide the readers with more context about the role and study of these RNA molecules in cancer. Also, the original figures 1 and 2 (now figures 3 and 4) were improved for clarity.

  1. What are the main questions to be addressed in the field?

Reply: We have added some questions in the introductory section's last paragraph to help the readers understand the actual state of knowledge about the functions of lncRNAs in VM in cancer (Page 3, lanes 101-107).

Often the notions presented here are very vague and generic. Then authors go on about enumerating a list of lncRNAs and mirRNAs per cancer type with a lot of repetitive wording that sounds quite empty at times. Some added value is needed compared to the current reviews already published in the topic.

Reply: Thanks for your pertinent suggestions. We have considered your comments and improved the writing by avoiding repetitive words. No similar reviews on lncRNAs and VM have been published in cancer; thus, the present review is valuable because it is the first report summarizing the increasing literature on this topic.

Some suggestions:

  1. The manuscript can be improved by adding comprehensive synopsis tables grouped by promoters and inhibitors of VM, adding information on the mechanism, etc, informed by a more expansive and integrative processing of the data retrieved in bibliography.

Reply: As the reviewer suggested, we have added tables 1 and 2, which grouped the lncRNAs that act as sponges and lncRNAs with functions other than sponges. The tables included cancer type, target microRNAs, and mechanisms of action. Also, we have added two new informative figures, 1 and 2, about the mechanisms of action of lncRNAs (Page 3) and a typical workflow for lncRNA characterization in cancer (Page 4) to provide the readers with more context about the role and study of lncRNAs in cancer. In addition, the original figures 1 and 2 (now figures 3 and 4) were improved for clarity.

  1. Identify and describe the main functions in the introduction before going in detail in each group (so far the main function described is as "miRNA sponges").

Reply: We have identified and described the main functions reported for lncRNAs in cancer, which are summarized in the new Figure 1. As the reviewer mentioned, the primary characterized function is miRNA sponges. However, other diverse mechanisms have been reported, including the degradation/stabilization of mRNAs and proteins, the recruitment of transcription factors to gene promoters, etc., which are delineated in Figure 1 (Page 3), tables 1 and 2, and in the corresponding introduction section (Page 2, lanes 73-89).

  1. Extract the commonalities and the cancer-specific trends.

Reply: As the reviewer suggested and according to the published evidence, several commonalities can be delineated for the mechanisms of action of lncRNAs during the VM process: i) they mainly act as sponges of miRNAs; ii) they function by establishing specific lncRNA/miRNA/mRNA axis, and iii) they modulate proteins involved in the ECM remodeling, e.g. metalloproteinases influencing cell migration and invasion. For instance, LINC00339 promotes cell proliferation, invasion, and VM by sponging miR-539-5p, resulting in TWIST1 depression, which, in turn, activates cell migration via metalloproteinases 2 and 14 (MMP2, MMP14) transcriptional activation. This discussion has been added to the revised version of the manuscript (Page 5, lanes 150-159).

  1. Which LncRNA are good candidates for RNA therapy? I was expecting a paragraph or table on that note mentioned initially.

Reply: Thank you very much for your wise observation. At the end of the manuscript, we added a section entitled "LncRNAs as therapeutic targets" and described some lncRNAs with potential as therapeutic targets that deserve pre-clinical considerations and/or evaluation in clinical trials.

  1. For clarity, I suggest present the figure's message with larger cartoons. The figures are very packed and the icons are too small.

Reply: We have modified figures 1 and 2 (now figures 2 and 3) and made the icons and letters bigger for better reader visualization.

  1. Work on more meaningful conclusions and mention briefly some future directions and open questions.

Reply: Thank you very much for pointing out this critical issue. As suggested, we have rewritten and improved the conclusion and proposed future perspectives.

Minor revision:

  1. Material and methods section seems unnecessary for describing a bibliography search, which is obvious in the context of a review paper.

Reply: We have deleted the materials and methods.

  1. Fix the typos in the figure number (all pointing to figure 1).

Reply: We have corrected the figure.

  1. Paragraph 3.5 on RCC: correct the typos "CRC" written many times.

Reply: The typos have been corrected.

  1. Do not abuse the statement "is poorly understood/characterized" (found 3 times in a single page).

Reply: Thanks for the comments. We have deleted the repetitive statement in several places in the revised manuscript. The manuscript's English grammar has been reviewed and corrected using the MDPI language services. The invoice is attached as supplementary data.

Reviewer 4 Report

Comments and Suggestions for Authors

Review of the manuscript ”LncRNAs regulate vasculogenic mimicry in human cancers” for the Cells journal.

The goal of this study was to reviewed the current knowledge about the role of lncRNAs in regulating vasculogenic mimicry (VM) in human cancers, with a special emphasis on their potential to be used as novel therapeutic targets in RNA-based molecular interventions. Authors have performed PubMed searches to identify the previously 84 reported lncRNAs associated with VM in cancer. The published manuscripts were classified and described according to the specific type of cancer (glioma, lung cancer, gastric cancer, breast cancer, renal cell carcinoma, osteosarcoma, colorectal cancer, hepatocellular carcinoma, sinonasal cancer). The findings of this study provide insights into the role of lncRNAs regulating VM in cancer. Data highlights the essential functions of these non-coding RNAs in modulating genes that activate VM in diverse types of neoplasia. Remarkably, lncRNA signatures from cancer patients can be used as novel molecular biomarkers of progression, prognosis, and VM. However, exhaustive functional validation of the lncRNAs as oncogenes or tumor suppressors to targeting with inhibitors or precursors using in vivo disease models is still needed.

However, during the review of this manuscript though, some remarks and comments appeared.

Minor comments:

  1. There are some spelling errors throughout the manuscript that should be carefully corrected during revision.
  2. In the text, references, Tables and Figures should not be written in bold font.
  3. The review article is extensive, but the number of cited works is rather small.

Author Response

Minor comments:

There are some spelling errors throughout the manuscript that should be carefully corrected during revision.

Reply: We would like to express our gratitude for your time and attention in reviewing our manuscript. We have carefully considered the comments and have made the necessary corrections, which we hope will meet with your approval. The English grammar of the manuscript has been reviewed and corrected using the MDPI language services. The certificate is attached as supplementary data.

In the text, references, Tables and Figures should not be written in bold font.

Reply: Thanks for the comments. As the reviewer suggested, we have deleted the bold fonts in the figures and tables in the revised manuscript. 

The review article is extensive, but the number of cited works is rather small.

Reply: Thanks for the comments. We have added several references to the manuscript to make the review more balanced. 

Round 2

Reviewer 1 Report

Comments and Suggestions for Authors

The manuscript has improved considerably. The authors addressed all issues raised previously. 

Author Response

Thanks to the reviewer for the positive opinion on our manuscript. We greatly appreciate your original comments, which improved the study's quality after reply.  

Reviewer 2 Report

Comments and Suggestions for Authors

This revised paper is quite good. I deem that it has met the publication requirements and there are no other problems.

Author Response

(The authors gave the same response as above.)

Reviewer 3 Report

Comments and Suggestions for Authors

After the extensive update of the manuscript, there has been a significant improvement in content and form. The message is more clear and informative. The figures have improve in visibility and clarity.

The new section of LnRNAs as therapeutic targets provides some introduction to the topic. Here, a table with a more exhaustive list of examples of current known FDA drugs that benefit from the combination with lnRNA inhibitors would be ideal (but optional at this point).

Minor editions:

Figure 4: missing a "VM" label in RCC row.

Italics for in vivo and in vitro along the manuscript.

Reference 60 do not have the same format as the others.

Author Response

After the extensive update of the manuscript, there has been a significant improvement in content and form. The message is more clear and informative. The figures have improve in visibility and clarity.

REPLY: We appreciate the reviewer's positive opinion on our manuscript. We also greatly appreciate your original comments, which improved the study's quality after you replied. 

The new section of LnRNAs as therapeutic targets provides some introduction to the topic. Here, a table with a more exhaustive list of examples of current known FDA drugs that benefit from the combination with lnRNA inhibitors would be ideal (but optional at this point).

REPLY: Thanks for the comments, it's a very good idea to include a list of examples of therapies based on lncRNAs intervention along with FDA-approved drugs. However, today, no examples of such combinations have been reported. We predict that more data about these combined therapies will be available in the following years, with particular success.

Minor editions:

Figure 4: missing a "VM" label in RCC row.

REPLY: Figure 4 was corrected. VM label was added to renal carcinoma panel.

Italics for in vivo and in vitro along the manuscript.

REPLY: Thanks for the suggestion. However, Cells MDPI format for in vitro/in vivo is without italics.

Reference 60 do not have the same format as the others.

REPLY: Reference has been corrected.